# Lipotransfer Strategies and Techniques to Achieve Successful Breast Reconstruction in the Radiated Breast

**DOI:** 10.3390/medicina56100516

**Published:** 2020-10-01

**Authors:** Kristina Crawford, Matthew Endara

**Affiliations:** 1Resident Physician, Vanderbilt University Medical Center, Nashville, TN 37232, USA; kristinaMcrawford@gmail.com; 2Plastic Surgeon, Maury Regional Medical Group, Columbia, TN 38401, USA

**Keywords:** lipotransfer, fat grafting, breast reconstruction, tissue expansion, radiotherapy, expander-to-implant, radiated breast

## Abstract

Radiation therapy is frequently a critical component of breast cancer care but carries with it side effects that are particularly damaging to reconstructive efforts. Autologous lipotransfer has the ability to improve radiated skin throughout the body due to the pluripotent stem cells and multiple growth factors transferred therein. The oncologic safety of lipotransfer to the breasts is demonstrated in the literature and is frequently considered an adjunctive procedure for improving the aesthetic outcomes of breast reconstruction. Using lipotransfer as an integral rather than adjunctive step in the reconstructive process for breast cancer patients requiring radiation results in improved complication rates equivalent to those of nonradiated breasts, expanding options in these otherwise complicated cases. Herein, we provide a detailed review of the cellular toxicity conferred by radiotherapy and describe at length our approach to autologous lipotransfer in radiated breasts.

## 1. Introduction

Reconstruction of the breasts following the surgical management of cancer is associated with improved quality of life, feelings of well-being, and psychosocial development [1,2]. The objective of the reconstructive surgeon should be to offer options that facilitate these goals while minimizing potential complications. Between 2004 and 2015, 2.4 million women were diagnosed with breast cancer [3]. Partial mastectomy was the most frequent surgical treatment and implant reconstruction was the most common reconstructive choice [3]. Acellular dermal matrix (ADM) is utilized in approximately 50% of breast reconstructions [4].

Radiation in the patient’s oncologic care is a well-known and well-studied risk factor for increased complications and reconstructive failure [5,6]. Since the appropriate management of cancer frequently requires radiation treatment (RT) to improve survival and recurrence rates, it is incumbent on the reconstructive surgeon to identify and implement strategies to compensate for this therapy. The traditional “gold standard” treatment for this has been the transfer of well-vascularized tissue in the form of a pedicled or free flap to reconstruct the resultant volume loss in the radiated breast. Though effective, these procedures may not always be available or the patient’s preferred option. Lack of access to properly trained reconstructive microsurgeons, inexperienced hospitals, and a paucity of donor sites impact patients’ ability to undergo these procedures. Patients may also wish to avoid procedures with potential donor site complications, increased operative time, requisite inpatient admission, or the longer postoperative recovery times that can be associated with these more complex surgeries. Autologous lipotransfer is a relatively simple procedure that is being increasingly recognized as a strategy in the radiated patient, with mounting evidence to support its use [7,8,9,10,11,12,13,14,15]. Though the ideal application of this technique remains debated in the literature, it is clearly becoming a critical and not simply adjunctive part of the reconstructive process in irradiated patients.

## 2. Use of Radiation in the Breast Cancer Patient

The use of RT in breast cancer patients irrefutably improves the survival and recurrence rates in lumpectomy and properly selected mastectomy patients [16,17,18,19]. Breast conserving therapy (BCT) as a treatment modality uses RT as a necessary step. As such, the vast majority of these patients are subject to RT as part of their treatment. The indications for use in mastectomy patients are expanding as well, with some centers offering it to as much as 70% of patients [19]. Indeed, a meta-analysis by the Early Breast Cancer Trialists Collaborative Group (EBCTCG) found improved rates for the 10-year recurrence and 20-year mortality in doing so [17]. The number of patients who require reconstruction and have been or will be exposed to RT is, therefore, increasing.

Irradiated tissues are associated with increased rates of surgical complication throughout the body [20,21]. This is especially true with radiated breast reconstruction, as evidenced by the higher rates of infection, capsular contracture, implant exposure, and overall reconstructive failure [22,23,24,25,26,27,28,29,30,31,32,33,34,35,36,37,38]. Though still the most common form of reconstruction being performed today, staged expander-to-implant-based reconstruction is especially sensitive to the unintended side effects of radiation. Interestingly, the timing of post-mastectomy radiotherapy may have a bearing on the complication rates [30,38]. Given the higher complication rate incurred by radiotherapy, some surgeons refuse to even offer an implant-based procedure to women who require radiation. Patients desiring procedures to correct asymmetries, ptosis, or macromastia following BCT can be at increased risk of complications such as delayed healing, prolonged edema, and breast loss necessitating flap reconstruction. Despite complications, these procedures are believed to be safe but with careful patient selection. One study found a pooled complication rate of 50% with mastopexy or a reduction in patients who had undergone BCT with RT [37]. Ultimately, an understanding of the harmful effect that radiation introduces to the surrounding tissues is paramount to success when designing treatment strategies for reconstruction in these patients.

The mechanisms by which RT is so effective at shrinking tumor size and local recurrence are the same ones that cause collateral side effects to local tissues. Radiation-induced tissue damage and the ensuing cellular and molecular response have been well-described in the literature [39,40,41,42,43,44]. The process occurs in three general phases: acute, latent, and late. The acute phase is thought to last from 0 to 6 months after exposure to radiation. This phase is characterized by damage to highly replicative cell lines through the initiation of cytokine cascades, the creation of reactive oxygen species, and the release of free radicals within the exposed cells. This property of RT is useful for causing apoptosis in cancer cells but is equally harmful to other proliferative cell lines such as basal keratinocytes and hair follicle stem cells. Damage to these regenerative cell lines results in the impairment of self-renewing abilities within the skin. Fibroblasts, endothelial cells, and epidermal cells within the radiation field are also affected, resulting in the release of a variety of molecular signals. This leads to the activation of the coagulation cascade, as well as increased inflammation, tissue remodeling, and epithelial regeneration. Finally, blood vessels, especially smaller arterioles and capillaries, are affected during this phase. These vessels demonstrate increased permeability and thus tissue edema, as well as the formation of fibrin plugs, with the resultant creation of local areas of ischemia.

The tissues proceed from the acute phase to a short latent period that is, as of yet, undefined but is believed to begin approximately 6 months after treatment [45,46]. The late-phase reactions occur next and can progress up to and beyond 20 years after initial exposure. The continued release of cytokines and growth factors results in prolonged fibroblast proliferation and progressive extracellular matrix deposition. Tissues become fibrotic, with a decrease in vascular density. These factors lead to sites inhospitable to surgical interventions, as they are stiff and have inadequate perfusion for healing. As such, these patients are frequently considered poor candidates for additional reconstructive procedures. Several studies have identified expander-to-implant surgeries as being particularly susceptible to these negative effects [23,25]. Patients desiring some form of breast reshaping after breast conservation therapy are equally approached with caution.

Strategies that take into account these harmful effects of radiation have been met with some success [47,48,49,50,51,52]. This includes delaying the expander-to-implant exchange procedure for 6 months to allow for the completion of the acute phase; using a counter incision in the IMF, thereby avoiding the more heavily radiated mastectomy incision line; and the use of autologous lipotransfer to physiologically reverse the harmful effects.

## 3. Autologous Lipotransfer to Regenerate Radiated Tissues

Though used for over 100 years to increase tissue bulk for cosmetic effect throughout the body, autologous lipotransfer is now seen as a particularly useful technique for treating radiodermatitis [53]. The reason it is so effective in this regard appears to be due to the multipotent adipose-derived stem cells, the adipose-derived regenerative cells transferred, and miscellaneous components of the stromal vascular fraction of the graft. It has been observed that adipose-derived stem cells (ASC) have the ability to regenerate new adipose tissue, ductal epithelium, and even nipple structures [54]. The mechanism by which adipose stem cells are capable of reversing the harmful effects of radiation is an area of active research [55]. It does appear that within radiated tissue the ASC is important, as it is capable of thriving and even proliferating in that ischemic environment [54]. Suspected mechanisms by which ASCs to promote the reversal of radiodermatitis are their ability to differentiate into lost cell types and to release paracrine signals with proangiogenic and anti-fibrotic effects.

Another factor that may contribute to the proangiogenic effect seen with lipotransfer into irradiated tissues is the inclusion of additional vessel-forming elements [56]. These include endothelial cells, pericytes, smooth muscle cells, and their progenitors capable of forming vascular cells and blood vessels. Experimental models transferring human fat to irradiated murine tissue has supported these findings. Indeed, the grafted tissue was found to have a decreased dermal thickness, reduction in collagen content, increase in vascular density, and overall improved fat graft retention.

Clinically, the beneficial effect of fat grafting in radiated patients has been demonstrated in several studies [8,9,10]. This technique for ameliorating the radiodermatitis of the breast is changing the way that reconstructive surgeons are approaching breast cancer patients. Initial concern over the potential for cancer activation limited the use of autologous lipotransfer in the breast. Multiple clinical studies, meta-analysis, and systematic reviews have failed to provide evidence to support this concern. Consequently, lipofilling had been used with increasing popularity in breast reconstruction, but typically as an adjunctive step to improve the final cosmetic result [57]. While oily cyst formation is a notable complication in a minority of patients, lipotransfer to the breast is generally regarded as a safe and well-tolerated procedure [58,59]. Moreover, the increased recognition of the positive effects on radiation tissue has resulted in the development of treatment protocols that incorporate it as an integral part of reconstructing these radiated patients.

## 4. Use of Autologous Lipotransfer in the Reconstruction of the Radiated Breast

Initial experience with lipotransfer in the radiated breast focused on using it to revive and prime post mastectomy skin flaps either after the completion of reconstruction or prior to attempting it [8,13,14,15]. These strategies were important for demonstrating efficacy in improving complication and failure rates but were limited in the cosmetic results they were able to obtain, delaying the overall time course.

Building on this, Ribuffo et al. presented 32 patients who underwent modified radical mastectomy followed by RT [7]. The patients were reconstructed in an immediate fashion at the time of mastectomy with the placement of tissue expanders in a submuscular plane. Half of the patients underwent between 1 and 2 separate autologous lipotransfer procedures as early as 6 weeks after the completion of radiotherapy before expander to implant exchange. They reported a 0% complication rate in their treatment arm and a 43% rate in the control group. Introducing lipotransfer as a separate but necessary part of their protocol was unique, and it became a formal part of their protocol for success.

Work by Serra-Renom et al. confirmed the utility of lipotransfer in 65 of their mastectomized irradiated patients by incorporating serial fat grafting into their protocol [9]. These patients underwent multiple fat grafting procedures, including before and at the time of expander to implant exchange. They found excellent clinical results with their technique. This study was limited, as the patients were not demonstrating significant acute effects of radiation in the form of radiodermatitis, and thus the severity of damage to the tissues was in question.

Our 3-stage lipo-approach to mastectomized irradiated patients is modeled on these previous studies and additional best available evidence for mitigating radiotoxicity (see Scheme 1). The hallmarks of our algorithm include the use of an ADM, maintenance of the expander in a fully inflated position during radiation, the delay of the expander-to-implant procedure for at least 6 months after radiotherapy completion, the use of a counter-incision at the inframammary fold (IMF) in cases of skin-sparing mastectomy (SSM), and the performance of a separate surgery whereby autologous fat is transferred to the radiated breast prior to the final exchange. Our algorithm is illustrated in Scheme 1. Comparing radiated breasts to our general non-radiated population as well as within for patients who had a bilateral mastectomy, whereby one breast was radiated and one was non-radiated, revealed equivalent complication rates (*p* = 0.387 and *p* = 1 respectively). Table 1 outlines our patient demographics. The clinical outcomes are detailed in Table 2 and Table 3.

A subset of patients who have undergone breast conservation therapy with lumpectomy followed by radiation desire either mastopexy or reduction to improve their postoperative appearance as well as improve the symmetry between their treated and untreated breast. The current recommendations are to limit offering these surgeries to carefully selected patients as long after radiation as possible. The use of fat grafting to “prime” the skin envelope as a separate procedure prior to attempting a reduction or a lift is an alternative strategy that has been successfully utilized in our practice (nonpublished) with reproducible and reliable results.

### 4.1. Three-Step Approach in the Mastectomy Patient Requiring Post Mastectomy Radiation

In radiation-naïve patients undergoing a mastectomy, the preoperative consultation includes a discussion of the three-stage approach to implant-based reconstruction, as well as the use of autologous flaps, should radiation be in question or required. Inherent in this approach is the use of flaps as a salvage procedure should implant failure arise. Patients who have already undergone radiation, such as previous breast conservation therapy patients now suffering a recurrence, are not candidates for this approach, and some form of autologous flap transfer is recommended.

Regardless of whether a patient would prefer flap or implant-based reconstruction, we prefer to proceed in a delayed immediate fashion. The placement of a tissue expander at the time of the mastectomy is therefore essential in this technique, precluding direct to implant or immediate flap reconstruction. Though evidence for the use of this technique applies to partial submuscular expander placement, it is currently being used for prepectoral reconstructions as well [60]. Cuomo et al. report better aesthetic results with prepectoral reconstruction [61]. In either case, the use of an acellular dermal matrix is considered an important part of the expander placement. The placement of this matrix creates a plane for ease of lipotransfer that is thought to be radioprotective.

The patient undergoes serial expansions starting two weeks after the index procedure. The breast to be radiated is expanded to the patient preference or the expander limit. On this side, we kept the tissue expander (TE) inflated during RT, as there is growing evidence that expander deflation leads to expander loss; skin dimpling; and the distortion of the inferior edge, leading to skin trauma upon re-inflation. [62,63]. An emerging study has also suggested decreased toxicity to the chest wall and underlying structures with the maintenance of TE inflation during RT [64]. In those who have undergone bilateral mastectomy, the non-radiated side is deflated prior to radiation, as this more effectively keeps this tissue out of the radiation field. The patient is monitored through their RT with a clinic visit at the halfway mark and following the completion of treatment. One week after the completion of radiation, the non-radiated breast is then easily re-inflated.

The patient is then subject to a 3-month waiting period prior to their next procedure: autologous lipotransfer to the radiated breast. At this point, the patient is taken back to the operating room for whole-breast fat grafting. This procedure is performed by utilizing a superwet technique for liposuction into a revolve fat transfer harvest system. Prior to the injection of the fat, pre-tunneling is performed within the subcutaneous space. The correct space is identified under direct visualization through a 1 cm incision within the mastectomy scar. The avoidance of using sharp tipped instruments is important during this step to avoid rupturing the expander. Scar tissue bands that may block the ability to uniformly inject the fat are severed using a riveted fat harvest cannula with a saw-type motion. The expander is then deflated by 60–100 mL to make room for the transfer of the fat. It is important to avoid over deflation, as the fat still requires a flat plane to be placed in a string of pearls fashion.

The fat is then processed with three washes of warm lactated ringers. It is transferred from the revolve to a 60 mL syringe, then into 3 mL syringes for transfer. The majority of the fat is injected through the incision. If additional access sites are needed to optimize the angle of delivery, they can be made with a 16-gauge needle while tenting the skin from the inside with a fat-grafting cannula to protect the device. Constant motion of the syringe while injecting the fat is critical to avoid the clumping of the fat graft and subsequent poor take. Enough fat is injected to fill the space by injecting at least as much as the fluid removed, but not to the point of skin discoloration or the creation of an overly taut skin envelope.

Following the completion of this step, the patient is monitored closely with weekly follow-up visits for the 3 weeks after surgery. If additional expansion is desired, this can be attempted at 1-month post fat grating. The patient is subject to another 3-month waiting period before their final surgery. This is timed to optimize the chances that the lipoaspirate will positively affect the acute phase of radiation injury while performing the surgery during the latent phase.

The patient is taken back to the operating room once more for the removal of the expander and the placement of the final implant. In cases of skin-sparing mastectomy, a counter-incision is utilized within the inframammary fold. Patients who have undergone a nipple-sparing mastectomy using an IMF incision are accessed by extending the incision laterally. Any requisite capsular modifications are able to be performed at this time to not only enhance the final shape of the reconstructed breast but also address any contraction that had occurred as a result of radiation. The implant is introduced using a touch-free technique. Closure is performed in layers using a buried, monofilament dissolvable suture. Please see patient Figure 1, Figure 2 and Figure 3.

### 4.2. Two-Stage Approach to BCT Patients Desiring Oncoplastic Mastopexy or Reduction After Completion of RT

Eligible patients typically present with severe asymmetry following the completion of their breast conservation therapy combined with ptosis and/or macromastia. These patients are counseled on the risks of operating in a previously radiated surgical field and the benefit of priming the tissues with autologous lipotransfer before doing so. A two-step approach is offered, with the first procedure consisting of a lift/reduction of the nonradiated breast and fat grafting to the radiated breast. The mastopexy/reduction of the radiated side is performed at least 3 months later.

The fat-grafting technique is similar as for mastectomy patients, with the preferred plane of injection remaining in the subcutaneous space. Care is taken not to inject directly into the breast tissue for multiple reasons. The subcutaneous plane is the target of transfer, as we are trying to reverse the negative radiation effects on those regenerative cell lines that, if damaged, will lead to increased risk in the final breast shaping procedure. Injection into the breast tissue would not accomplish this goal and, though not proven by data, would be more concerning from an oncologic perspective. Multiple patients have been operated on in this manner by the primary author, with the largest complication being persistent asymmetry that was still improved from before surgery. The technique has allowed for a reduction in the selectivity for offering breast reshaping in previously radiated fields, as has been recommended by previous authors.

## 5. Conclusions

The recognition of the regenerative properties of lipoaspirate in radiated fields is leading to the simplification of breast reconstruction in this otherwise complicated patient population. The optimal strategy for this requires knowledge of the harmful effects of radiation, their time course, and the biomolecular pathways by which lipoaspirate reverses them. Though not yet fully understood, it is clear that applying autologous lipotransfer can significantly improve breast reconstruction outcomes in irradiated patients. The ease of these fat-grafting procedures along with their use and application throughout the body makes them very well-known to most plastic surgeons and may be leading to a paradigm shift in approaching the radiated breast. This has the potential to improve options and access to reconstruction for this ever-growing patient group.

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
