# Peer review of "Lipotransfer Strategies and Techniques to Achieve Successful Breast Reconstruction in the Radiated Breast"

_medicina, 2020, doi:10.3390/medicina56100516_

Round 1
Reviewer 1 Report
This is an important aspect in the management of breast cancer and contributes to try and achieve a good cosmetic outcome following the surgery and especially after radiotherapy.
DIEP flap is the gold standard for reconstruction in an irradiated post mastectomy patient. The advent of ADM has helped but still has varying results post DXRT. The fat grafting technique may help in this group of patients who opt to have implant with ADM and do not have the DIEP reconstruction due to either personal choice or because of the patient characterstics which make them not suitable for this.
Author Response
Thank you so much for reviewing our article. We appreciate your time.
Reviewer 2 Report
The purpose of this article is very interesting.
Characteristics of staminal cells, physiologically present in fat tissue, are not well defined in the present study (line 12 vs line 101).
The treatment algorithm is not well defined. Please explain why the lipofilling sessions are performed 3 months after radiotherapy and not before, as suggested by Ribuffo et al.
Please report evidences on the process of TE deflation with the aim of reducing hearth and lung toxicity and explain why you perform it only on non-tumor breasts.
Please, better define the “internal controls” in Table 3, it is not clear if these controls are bilateral mastectomy cases (omolateral cancer vs contralateral profilaxis).
At page 7, picture c is missing in Figure 1. Moreover, pictures are not suitable for representing breast reconstruction cases results (in picture “a” the patient is slightly rotated, the umbilicus is not exposed and picture dimensions are not correct, in picture “b” the patient’s arms are standing upward). Please provide valid photographic data.
Discussion is missing, expounding your results a comparison with literature should underline the benefits of your algorythm.
Author Response
“Characteristics of staminal cells, physiologically present in fat tissue, are not well defined in the present study (line 12 vs line 101).”
We appreciate this comment and agree that we have not extensively expounded stem cells specifically. However, defining stem cells is not the central focus of our study. Instead, we include cellular/molecular processes that are implicated in radiation toxicity and reference literature that discusses in more detail adipose-derived stem cells (Lines 65-80 and Lines 101-115 in revised manuscript. See referenced papers by Rigotti, Hong, Hymes, Hauer-Jensen, Lopez).
“The treatment algorithm is not well defined. Please explain why the lipofilling sessions are performed 3 months after radiotherapy and not before, as suggested by Ribuffo et al.”
Waiting three months s/p RT completion allowed lipofilling to occur within the acute phase while avoiding some of the tissue effects (e.g. edema) customarily seen in the immediate post-RT period. Three months also permitted adequate time for bilateral mastectomy patients to undergo expander inflation of their un-radiated side. In addition, research out of UCSF has identified 6 months as being the earliest time point in which the complication rate begins to decrease for expander-to-implant reconstruction after RT. By choosing 3 months after RT, we are able to add yet another factor to decrease complication rate while not delaying their ability to undergo exchange within this time period.
“Please report evidences on the process of TE deflation with the aim of reducing hearth and lung toxicity and explain why you perform it only on non-tumor breasts.”
Deflating the expander in the non-tumor breast reduces the chance of the TE being within the radiation field. For the tumor breast, we kept the TE inflated during RT as there is growing evidence that expander deflation leads to expander loss, skin dimpling, and distortion of the inferior edge leading to skin trauma upon re-inflation. An emerging study has also suggested decreased toxicity to the chest wall and underlying structures by leaving the device inflated. These citations were added to the manuscript (please see referenced literature by Woo, Celet, and Amro).
“Please, better define the “internal controls” in Table 3, it is not clear if these controls are bilateral mastectomy cases (omolateral cancer vs contralateral profilaxis).”
“Internal controls” were patients who had one radiated breast and one non-radiated breast. As such, these patients could serve as their own control counterpart. No difference in complication by radiotherapy exposure was found. Table 3 has been updated to better define this group.
“At page 7, picture c is missing in Figure 1. Moreover, pictures are not suitable for representing breast reconstruction cases results (in picture “a” the patient is slightly rotated, the umbilicus is not exposed and picture dimensions are not correct, in picture “b” the patient’s arms are standing upward). Please provide valid photographic data.”
Picture 1c has been added. We agree that in Picture A the patient is indeed slightly rotated. However, unfortunately, the photos cannot be re-taken.
“Discussion is missing, expounding your results a comparison with literature should underline the benefits of your algorithm.”
With this being a review article, we do not have a formal discussion section as we have integrated the discussion throughout the paper. The first half of the paper discusses in-detail the use of radiation and its associated complications, the effects of radiotoxicity and stages of tissue damage, as well as existing research on autologous lipotransfer. We have included sub-headings to better define these sections. We have organized the paper in this manner in order to set the stage for how we developed our algorithm. We are open to re-organizing the paper by moving the discussion to the end, however we believe the existing format effectively introduces the reader to established studies that help explain both the rationale and conception of our algorithm. The benefit of our algorithm is supported by our postoperative outcomes which is detailed in our tables and figures.
